# Factors and determinants associated with prevalence of stunting and thinness among adolescents of Tharparkar, Sindh, Pakistan: A community-based study

**Aisha Memon, Zulfiqar Ali Laghari, Ayaz Ali Samo**(ID)*

Department of Physiology, Faculty of Natural Sciences, University of Sindh, Jamshoro, Sindh, Pakistan

* ayazsamo@usindh.edu.pk

## Abstract

Stunting and thinness are significant public health concerns for developing countries, including Pakistan. Previously conducted studies in Tharparkar have focused on stunting, and thinness, in children, and micronutrient deficiency, However, there was a paucity of literature on prevalence rates and risk factors associated with stunting and thinness among adolescents in Tharparkar, Sindh, Pakistan. This study aimed to assess the demographic and dietary factors related to the prevalence of stunting and thinness among adolescents in Tharparkar. A community-based cross-sectional study was conducted from June 2022 to August 2022. Most populated Union councils of all sub-districts of Tharparkar district were selected for sampling. A multi-stage clustering sampling strategy was used. Healthy adolescents with ages≥10 to ≤19 years were included in the study. Anthropometry was performed using standard methods. Stunting and thinness were defined as per WHO criteria. Dietary assessments were conducted on a weekly recall basis. Mean Frequency and chi-square were computed using SPSS software. The overall mean age of the study participants was 14.11±2.43 years, and the mean weight was 36.842±8.83 kgs. The mean height of the study participants was 149.0151±11.27 centimeters. The mean height for age z score was -1.3094±1.17. The mean body mass index for age z score was -1.5473±1.27. Among 599 participants the overall prevalence of stunting and thinness was 26.7% and 35% respectively. Risk factors such as late adolescence ($\chi2$=10.55 p=0.005), illiteracy, and less education ($\chi2$=8.41 p=0.03), the rural area ($\chi2$=3.92 p=0.04) significantly associated with an increased prevalence of stunting. The risk factors such as male ($\chi2$=13.11 p<0.0001), infrequent consumption of eggs ($\chi2$=5.2 p=0.02), and infrequent consumption of fresh vegetables ($\chi2$=5.2 p=0.02) were associated with an increased prevalence of thinness. This study underscores the urgent need for comprehensive nutritional programs that could focus on vulnerable demographic groups. Interventions should focus on improving dietary intake.

## Background

Nutrition plays an important role in an individual's physical and mental growth. The body's demand for energy increases with age. Adequate nutrition from childhood to

**Data availability statement:** All relevant data are within the paper and its Supporting Information files.

**Funding:** This study was funded by the research grant received from the Higher Education Commission, Islamabad through the National Research Program for Universities (NRPU) project number 14406 for 2021-2024.

**Competing interests:** The authors have declared that no competing interests exist.

adulthood is critical for performance of physiological functions. Adolescence (11–19 years) is a critical age period in which an individual transitions from childhood to adulthood. The significant changes include growth, increases in height, and weight, hormonal changes, the appearance of secondary sexual characteristics, changes in voice, and cognitive development [1]. Nutritional availability and affordability greatly vary around the globe due to wealth accumulation by 1% of rich people and widening economic inequalities [2]. Most people who live in low- and middle-income countries face problems including poverty, illiteracy, inflation, low income, unemployment, and intra-generational nutritional issues such as stunting and thinness from childhood to adolescence. Stunting and thinness affect poverty-ridden people irrespective of gender. Adolescent females in the low and middle-income South Asian countries are no exception. United National International Children Emergency Fund (UNICEF) the prevalence of stunting and thinness among South Asian adolescent females was 11% and 39% respectively. South Asian adolescent females were consuming a poor diet that did not meet the dietary requirements for their physical growth. Stunting and thinness were more common in female adolescents from rural areas, large families, and unskilled, uneducated parents with low-income households [3]. Though South Asian females are more vulnerable to stunting and thinness adolescent males are also affected by stunting and thinness. The prevalence of stunting and thinness among teenage boys in Uttar Pradesh, India was 25.6% and 25.8% [4]. The prevalence of stunting and thinness among male adolescents in Chowhali, Bangladesh was 46.6% and 42.4% [5].

The population of Pakistan is 241 million with a growth rate of 2.55% [6]. The country's gross domestic product (GDP) to debt ratio is 74.4% [7]. The World Health Organization has recommended that each country should spend at least 6% of the total GDP on the health of its people. However, last year the federal government of Pakistan allocated only PKR 24.25 billion for the health sector which is 2.8% of the total development budget and 0.05% of the GDP [8]. According to the Multidimensional Poverty Index (MPI) Survey in Pakistan, Balochistan has the highest level of multidimensional poverty, with 70% of its population considered poor. Khyber Pakhtunkhwa (KP) follows with 48%, and Sindh with 45%. Punjab has the lowest rate of multidimensional poverty at 30%, which is also below the national average of 39.1%. Over recent years, the levels of multidimensional poverty have generally decreased in most provinces except for Sindh. For example, Balochistan dropped slightly from 72.4% in 2014–15 to 70.5% in 2019–20. In KP, it decreased from 49.1% to 48.8%. In Punjab, it went down from 31.0% to 30.4%. However, in Sindh, multidimensional poverty increased from 43.1% in 2014–15 to 45.2% in 2019–20 [9]. Adolescents in such socio-economic conditions face several challenges contributing tostunting and thinness. Such challenges include food insecurity, low income, large families, lack of education, and poor healthcare.

Previously conducted studies from various areas of Pakistan have focused on the prevalence of stunting and thinness among adolescents in either urban [10] or semi-urban areas, female adolescents [11], and late adolescents [12], Tharparkar is the deserted area with the highest rates of mortality in children under five years of age, previously conducted studies in Tharparkar have focused on stunting, thinness, and wasting in children under 5 years of age [13], school-aged children [14] micronutrient deficiency [15,16]. However, no study has identified the sociodemographic and dietary factors associated with stunting and thinness among adolescents in Tharparkar, Sindh, Pakistan. The objective of the study was to assess the sociodemographic and dietary factors associated with the prevalence of stunting and thinness among adolescents in Tharparkar.

## Methodology

### Study area

Tharparkar district is a deserted area by topography. Tharparkar has 7 Tahsils (sub-districts) including Mithi, Islamkot, Diplo, Kaloi, Chachro, Dahli, and Nangar Parkar. The vegetation, desert tourism, small businesses, and livestock are the main sources of livelihood. The region is affected by climate change. Poverty, hunger, low economic status, access to healthcare facilities and inadequate health facilities are common problems for people of Tharparkar. Poverty, Illiteracy, and lack of awareness of diet and its role in preventing thinness and stunting and their health consequences make the adolescent population vulnerable to nutrition inadequacy-related problems.

### Study setting

A survey-based cross-sectional study was conducted to assess the demographic and dietary risk factors associated with the prevalence of stunting and thinness among adolescents in Tharparkar District's selected areas from June 2022 to August 2022. Data was collected using a multistage cluster sampling method. To ensure representation, the most populated villages from each sub-districts most populated Union councils were targeted. Within these selected villages, households were chosen randomly to participate in the survey. Subsequently, within the randomly selected households, adolescents were chosen for inclusion in the survey using a lottery method. This approach further added to the randomness of the selection process. The multistage cluster sampling method was designed to provide a comprehensive and representative sample of adolescents in Tharparkar for the survey.

### Sample size

The sample size was calculated using the following formula Sample Size (n) = [(Z^2) * p * (1 - p)]/ (E^2) Where: n: Sample size, - Z: Z-score corresponding to the desired confidence level approximately 1.96 for 95% confidence), p: Expected stunting prevalence (as a decimal) - E: Margin of Error (as a decimal). Because there is no previous literature on the prevalence of stunting and thinness among adolescents of Tharparkar. So, the average national prevalence of stunting and thinness of 25% percent was taken. Twenty-five percent expected prevalence (0.25), a 95% confidence level, and a 5% margin of error, as a result, a sample size of approximately 424 adolescents was needed for the study. We collected more than 600 samples. A total of 635 samples were collected. However, 36 samples were excluded for various reasons: age above 19 [4], age below 10 [4], married individuals [12], recent diarrhea cases [6], and outliers [10]. This left 599 samples for analysis. (Table 1).

### Informed consent

Most of the participants were under 18 years old, after explaining the research project objectives to their parents, written informed consent for data collection was obtained from parents.

### Data collection

A team of trained data collectors was formed. The team was trained to collect data in a rural setting under the supervision of a trained Ph.D. student. All those participants whose permission was obtained were asked to go to station 1 for an interview about sociodemographic and dietary factors, then station 2 for anthropometry.

**Table 1. Sample flow chart.**

| Tharpakar District | | | | | | | |
|---|---|---|---|---|---|---|---|
| Sub-Districtr | Nangar Parkar | Mithi | Chhachhro | Kaloi | Islamkot | Diplo | Dahli |
| Union Council | Nangar Parkar | Mithi | Chhachhro | Kaloi | Islamkot | Diplo | Dahli |
| Union Council | Satidera | Chelhar | Kantio | Bhitato | Sengahro | Jhirmiryo | Laplo |
| Union Council | Verawah | M.Veena | Mithrio Charan | Khetlari | Khario | Sobhiyar | Parno |
| Union Council | | | Rajoro | | | | |
| Samples# | n=90 | n=100 | n=92 | n=88 | n=89 | n=91 | n=85 |

| | | |
|---|---|---|
| Total Samples = 635 | |
| Samples Excluded | |
| Age>19=4 | |
| Age<10=4 | |
| Married=12 | |
| Diarrhea last 15 days=6 | |
| Outliers=10 | |
| Samples included = 599 | |

## Variable setting

The presence or absence of stunting and thinness were considered dependent variables. Independent variables included Demographic factors such as age, gender, and ethnicity, socioeconomic factors such as education level, parental education, siblings, income, and total family members, dietary factors such as breakfast, lunch, dinner, and type of diet, Height for age Z score (HAZ) above 0 was considered normal between -2 and -0.99 is considered mildly stunted, HAZ between -2 and -3 was considered Moderately stunted HAZ less than -3 was considered severely stunted. Body Mass Index for age Z score (BAZ) above 0 was considered normal between -2 and -0.99 is considered mild, BAZ between -2 and -3 was considered Moderately thin BAZ less than -3 was considered severely thin.

## Anthropometry

Participants were asked to remove heavy clothes, shoes, and other accessories. Weight and height were measured using calibrated scales placed on level surfaces. Weight and height were recorded to the nearest 0.1 kilograms and 0.5 inches, respectively. Height for age Z score (HAZ) and Body Mass Index for age Z score (BAZ) were calculated using the World Health Organization (WHO) recommended Anthro plus software.

**Inclusion and exclusion criteria.** Healthy unmarried adolescents who permanently belong to district Tharparkar without any disease history in the past 3 months were included in the study. Unhealthy, married, under or over the age of adolescence with any disease history in the past 3 months were excluded from this study.

## Ethical approval

Ethical approval was obtained from the Research Ethical Review Committee of the University of Sindh, Jamshoro.

## Statistical analysis

Special Package for Social Sciences (SPSS) software version 23 was used to compute frequencies, mean, SD, and chi-square and binary logistic regression analysis. The significance was determined using a P value of 0.05. WHO Anthro Plus software was used to calculate the Z score.

## Results

### General characteristics of study participants

The overall mean age of the study participants was 14.11±2.43 years, and the mean weight was 36.842±8.83 kgs. The mean height of the study participants was 149.0151±11.27 centimeters. The mean height for age z score was -1.3094±1.17. The mean body mass index for age z score was -1.5473±1.27. The gender-wise mean values are shown in Table 2.

### Basic demographic characteristics of the study participants

The demographic analysis revealed, 50% of the adolescents were of the age of early adolescence, 51.3% of the adolescents were females, 71.3% of the adolescents were literate, 56% of the adolescents had literate fathers and 86% of the adolescents had illiterate mothers, 87.4% of the adolescent's fathers had income ≤33000 Pakistani Rupees (PKR), 62.5% of the adolescents had >7family members and 75% of the adolescents had ≤7 siblings. (Table 3).

### Prevalence of stunting and its associated demographic and dietary factors

The overall prevalence of stunting was 27% among adolescents of Tharpakar. Twenty-eight percent of females and 26% of males were stunted. (Table 4 and Figs 1 and 2)

The late adolescence group was significantly associated with an increased prevalence of stunting ($\chi^2$=10.55 $p$=0.005). Illiterate and less educated groups (education ≤5 class) were associated with an increased prevalence of stunting ($\chi^2$=8.41 $p$=0.03). The rural area group was significantly associated with an increased prevalence of stunting ($\chi^2$=3.92 $p$=0.04) (Table 5). Dietary parameters are given in S1 Table. A binary logistic regression model examined the association between stunting and demographic factors. Primary and higher secondary education levels and middle adolescence stage are significantly associated with reduced odds of stunting (Table 6).

### Prevalence of thinness and its associated demographic and dietary factors

The overall prevalence of thinness was 35% among adolescents of Tharpakar. Twenty-eight percent of females and 42% of males were thin. (Table 7 and Figs 3 and 4).

The male group was significantly associated with an increased prevalence of thinness ($\chi^2$=13.11 $p$<0.0001). The literate group was associated with an increased prevalence of thinness ($\chi^2$=12.04 $p$=0.007). (Table 8) Infrequent consumption of eggs was associated with an increased prevalence of thinness ($\chi^2$=5.2 $p$=0.02). Infrequent consumption of fresh vegetables was associated with an increased prevalence of thinness ($\chi^2$=5.2 $p$=0.02). (Table 9). A binary logistic regression model examined the association between thinness and demographic factors. Gender (being male) and primary education are significant predictors of thinness. Males are at higher odds for thinness, while individuals with primary education have significantly lower odds of thinness compared to those who are illiterate (Table 10).

**Table 2. General characteristics of Study participants.**

| General Characteristics | Overall (599) | Female (n=307) | Male (n=292) |
|---|---|---|---|
| Age in years | 14.1134 ±2.43 | 14.3455±2.47 | 13.8693±2.37 |
| Weight (kg) | 36.842±8.83 | 36.820±7.23 | 36.866±10.26 |
| Height (cm) | 149.0151±11.27 | 147.6104±7.47 | 150.4920±14.07 |
| HAZ | -1.3094±1.17 | -1.3840±1.08 | -1.2311±1.26 |
| BAZ | -1.5473±1.27 | -1.3360±1.16 | -1.7696±1.35 |

**Table 3. Basic demographic characteristics of study participants.**

| Characteristics. | Frequency | Percentage (%) |
|---|---|---|
| **Age** | | |
| Early | 299 | 49.9 |
| Middle | 190 | 31.7 |
| Late | 110 | 18.4 |
| **Gender** | | |
| Female | 307 | 51.3 |
| Male | 292 | 48.7 |
| **Area** | | |
| Rural | 490 | 81.8 |
| Urban | 109 | 18.2 |
| **Education** | | |
| Illiterate (0) | 172 | 28.7 |
| Primary (1–5) | 137 | 22.9 |
| Secondary (6–10) | 248 | 41.4 |
| Higher Secondary (>10) | 42 | 7 |
| **Father's Education** | | |
| Illiterate | 260 | 43.4 |
| Literate | 339 | 56.6 |
| **Mother's Education** | | |
| Illiterate | 516 | 86.1 |
| Literate | 83 | 13.9 |
| **Socioeconomic status** | | |
| Monthly income ≤33000PKR | 503 | 84.0 |
| Monthly income ≥33100PKR | 95 | 15.9 |
| **Siblings** | | |
| < 7 | 451 | 75.3 |
| >7 | 148 | 24.7 |
| **Family Members** | | |
| < 7 | 227 | 37.9 |
| >7 | 372 | 62.1 |

## Discussion

This study aimed to assess the prevalence and demographic factors associated with stunting and thinness among adolescents of Tharparkar Sindh. The overall prevalence of stunting and thinness was 27%, and 35% respectively. Key risk factors identified included age, gender, education, and dietary factors such as chicken, egg, and fresh vegetables. Our findings indicated that the overall prevalence of stunting falls within the range reported in previous research, which has documented rates from 22.72% [17] to 37% [10]. Regarding thinness, the prevalence of thinness slightly exceeds the range reported in previous research, which documented rates from 9.3 [12] to 31% [10].

In the context of gender differences, our results indicated that the prevalence of stunting is nearly similar between females 28%, and males 25% while the prevalence of thinness showed high variance with 42% in males and 28% in females, indicating potential disparity between the genders. This disparity has also been noted in other studies conducted in Nigeria, Ethiopia, and India [18–20]. This could be because of the increased energy demands of males

**Table 4. Overall and gender-wise prevalence of stunting and its categories according to Z-score 2007 WHO reference.**

| Stunting | Frequency | Percentage |
|---|---|---|
| **Overall** | | |
| Normal ≤0. Z-score | 229 | 38.2% |
| Mild ≤1. Z-score | 210 | 35.1% |
| Moderate ≤2. Z-score | 125 | 20.9% |
| Severe ≤ 3. Z-score | 35 | 5.8% |
| **Female** | | |
| Normal ≤0. Z-score | 103 | 33.6% |
| Mild ≤1. Z-score | 117 | 38.1% |
| Moderate ≤2. Z-score | 74 | 24.1% |
| Severe ≤ 3. Z-score | 13 | 4.2% |
| **Male** | | |
| Normal ≤0. Z-score | 126 | 43.2% |
| Mild ≤1. Z-score | 93 | 31.8% |
| Moderate ≤2. Z-score | 51 | 17.5% |
| Severe ≤ 3. Z-score | 22 | 7.5% |

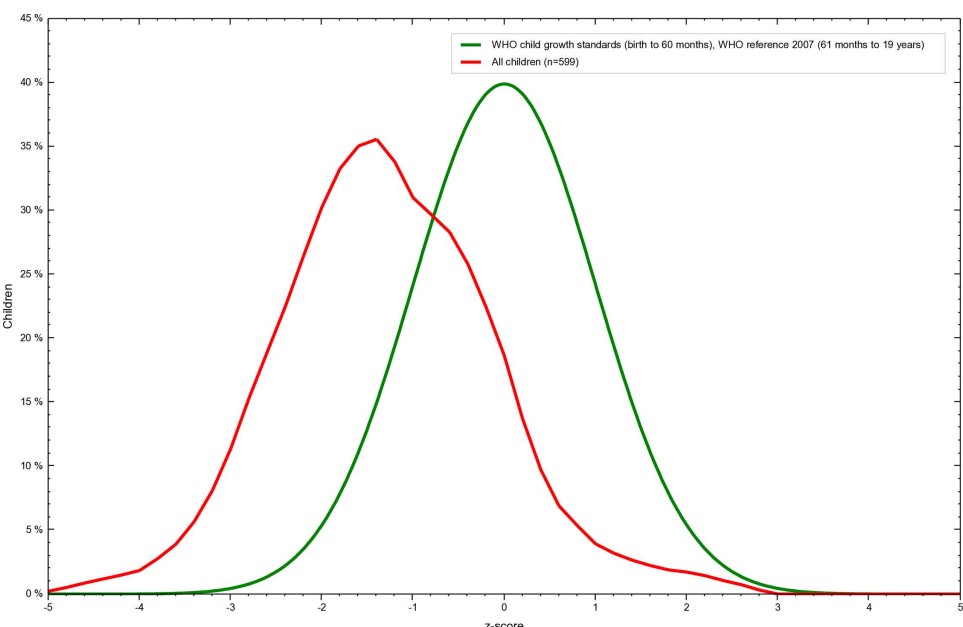

**Fig 1. Overall Z-score distribution of stunting among adolescents of Tharparkar, Sindh, Pakistan.** "This graph illustrates the overall Z-score distribution of height-for-age (stunting) among Adolescents of Tharparkar, Sindh, Pakistan generated using WHO AnthroPlus software. The Z-scores are calculated based on the 2007 WHO Growth Reference, with values below -2 standard deviations indicating stunting. The figure highlights the prevalence of malnutrition in the studied population and visually represents their growth and nutritional status relative to global standards.

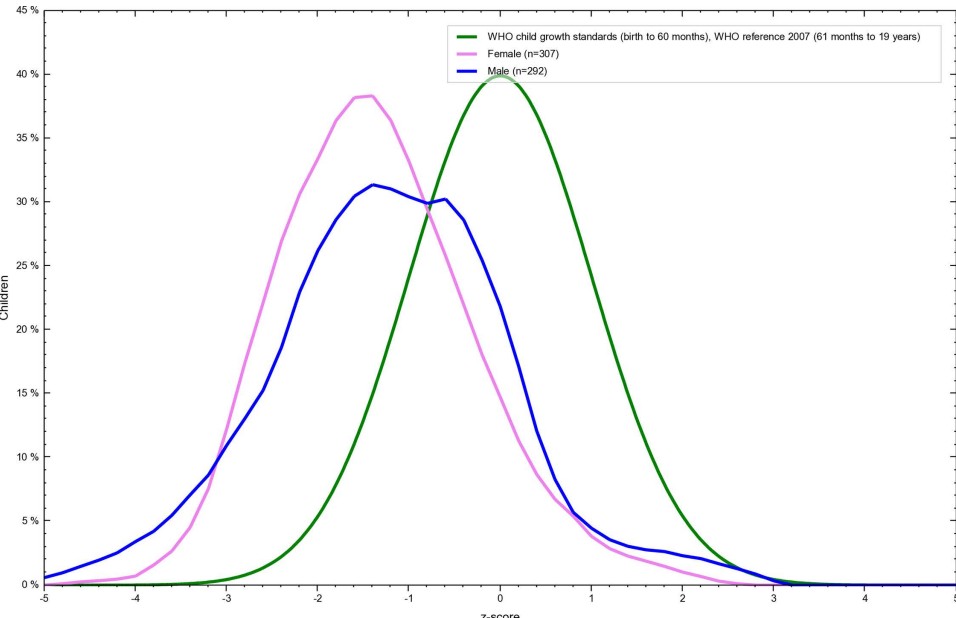

**Fig 2. Gender wise Z-score distribution of stunting among adolescents of Tharparkar, Sindh, Pakistan.** This graph illustrates the genderwise Z-score distribution of height-for-age (stunting) among Adolescents of Tharparkar, Sindh, Pakistan generated using WHO AnthroPlus software. The Z-scores are calculated based on the 2007 WHO Growth Reference, with values below -2 standard deviations indicating stunting. The figure highlights the prevalence of malnutrition in the studied population and visually represents their growth and nutritional status relative to global standards.".

which are not fulfilled compared to females and socio-economic and cultural factors could be attributed to explain this problem.

Demographic and dietary factors are important in explaining disparities in the prevalence of stunting and thinness. Our findings indicated that factors such as middle and late adolescent age group, rural area group, illiterate, and less educated group (participants with education up to class 5) were positively associated with increased prevalence of stunting among adolescents. Our findings are in line with previous studies [11,19].Our results showed that the male and educated groups were associated with an increased prevalence of thinness. Analysis of dietary factors showed that infrequent consumption of eggs and fresh vegetables was significantly associated with an increased prevalence of thinness although infrequent consumption of chicken did not show a statistically significant association, it showed a trend towards increased thinness. The findings of this study about the increased prevalence of thinness among males are consistent with other studies [18,19]. Several previous studies have shown that the illiterate group is a risk factor for the increased prevalence of thinness [21,22] Our findings suggest that an educated group is associated with an increased prevalence of thinness. Our finding could be due to the sample sizes in the last two educational classes (particularly the group with education more than the 10th class) being relatively small. The small sample size may lead to statistical variability. The study has limitations such as the small representation of the late adolescent age group, potential recall bias in dietary assessments, and causal inferences because of the cross-sectional study. These limitations must be considered in future studies.

The high prevalence of stunting and thinness among adolescents in our study underscores the urgent need for comprehensive nutritional programs that could focus on the demographic

**Table 5. Association of demographic factors with prevalence of stunting among adolescents of Tharparkar. Sindh, Pakistan.**

| demographic Factors | Stunting No | Stunting Yes | Chi-squire | P-value |
|---|---|---|---|---|
| **Age** | | | | |
| Early | 236 (78.9%) | 63(21.1%) | 10.558 | .005 |
| Middle | 130 (68.4%) | 60(31.6%) | | |
| Late | 72 (65.5%) | 38(34.5%) | | |
| **Gender** | | | | |
| Female | 220 (71.7%) | 87(28.3%) | .684[a] | .408 |
| Male | 218 (74.7%) | 74(25.3%) | | |
| **Area** | | | | |
| Rural | 350 (71.4%) | 140(28.6%) | 3.928 | .047 |
| Urban | 88 (80.7%) | 21(19.3%) | | |
| **Education** | | | | |
| Illiterate (0) | 119 (69.2%) | 53 (30.8%) | | |
| Primary (1–5) | 109 (76.6%) | 28 (20.4%) | 8.41 | 0.03 |
| Secondary (6–10) | 174 (70.2%) | 74 (29.8%) | | |
| Higher Secondary (>10) | 36 (85.7) | 6 (14.3%) | | |
| **Father's Education** | | | | |
| Illiterate | 185 (71.2%) | 75 (28.8%) | .905 | .341 |
| Literate | 253 (74.6%) | 86 (25.4%) | | |
| **Mother's Education** | | | | |
| Illiterate | 375 (72.7%) | 141 (27.3%) | .379 | .538 |
| Literate | 63 (75.9%) | 20 (24.1%) | | |
| **Socioeconomic status** | | | | |
| Monthly income ≤33000PKR | 329 (65.3%) | 175 (34.7%) | .040 | .841 |
| Monthly income ≥33100PKR | 61 (64.2%) | 34 (35.8%) | | |
| **Siblings** | | | | |
| < 7 | 331 (73.4%) | 120 (26.6%) | .068 | .794 |
| >7. | 107 (72.3%) | 41 (27.7%) | | |
| **Family Members** | | | | |
| < 7 | 171 (75.3%) | 56 (24.7%) | .907 | .341 |
| >7 | 267 (71.8%) | 105 (28.2%) | | |

groups identified by this study. Interventions should focus on improving dietary diversity and addressing socioeconomic disparities.

In conclusion, this study highlights significant prevalence rates of stunting and thinness among adolescents and identifies critical risk factors. Addressing these issues through targeted public health interventions is essential for improving adolescent health and preventing long-term adverse outcomes. The Federal Government of Pakistan and the Provincial Government of Sindh should allocate resources to start nutrition programs in schools and communities for the adolescent population to prevent stunting and thinness among the adolescent population of Tharparkar.

## Supporting information

**S1 Table. Association of dietary factors with prevalence of stunting among adolescents of Tharparkar, Sindh, Pakistan.**
(DOCX)

**Table 6. Binary logistic regression analysis stunting adjusted with demographic factors.**

| Variables in the Equation | | B | S.E. | Wald | df | Sig. | Exp(B) | 95% C.I.for EXP(B) | |
|---|---|---|---|---|---|---|---|---|---|
| | | | | | | | | Lower | Upper |
| Step 1[a] | Gender (1) | -.096 | .210 | .210 | 1 | .647 | .908 | .602 | 1.370 |
| | Area (1) | .529 | .280 | 3.578 | 1 | .059 | 1.697 | .981 | 2.936 |
| | Illiterate | | | 9.003 | 3 | .029 | | | |
| | Primary Education | 1.177 | .504 | 5.454 | 1 | .020 | 3.244 | 1.208 | 8.712 |
| | Secondary Education | .919 | .524 | 3.079 | 1 | .079 | 2.508 | .898 | 7.003 |
| | Higher Secondary Education | 1.343 | .493 | 7.430 | 1 | .006 | 3.830 | 1.458 | 10.059 |
| | Family Members (1) | -.148 | .216 | .468 | 1 | .494 | .862 | .564 | 1.318 |
| | Siblings (1) | .057 | .238 | .058 | 1 | .809 | 1.059 | .665 | 1.687 |
| | Early Adolescence | | | 13.793 | 2 | .001 | | | |
| | Middle Adolescence | -.951 | .268 | 12.554 | 1 | .000 | .386 | .228 | .654 |
| | Late Adolescence | -.426 | .274 | 2.419 | 1 | .120 | .653 | .382 | 1.117 |
| | Constant | -1.925 | .514 | 14.042 | 1 | .000 | .146 | | |

[a]Variable(s) entered on step 1: Gender (1), Area (1), Education Level (Illiterate, Primary, Secondary, Higher Secondary), Family Members (1), Siblings (1), Adolescence Stage (Early, Middle, Late)

Outcome **variable**: Thinness status (0 = Not Stunted, 1 = Stunted)

Predictor **variables**: **Age**: Early, Middle and Late (Reference Category: Late), **Gender**: Male or Female (reference category = Male), **Socioeconomic status**: Categorical, coded as low, middle, high (reference category = High), **Education level**: Primary, Secondary, Higher (reference category = Higher), **Urban/Rural residence**: Place of residence (Urban = 1, Rural = 0; reference =Urban) Adjusted **odds ratio (OR)**: Odds ratio adjusted for age, gender, socioeconomic status, education level, and place of residence. 95**% Confidence Interval (CI)**: The range within which the true odds ratio is expected to fall, with 95% certainty.

**Table 7. Overall and gender-wise prevalence of thinness and its categories according to Z-score 2007 WHO reference.**

| Thinness | Frequency | Percentage |
|---|---|---|
| **Overall** | | |
| Normal ≤0. Z-score | 198 | 33.1% |
| Mild ≤1. Z-score | 192 | 32.1% |
| Moderate ≤2. Z-score | 126 | 21.0% |
| Severe ≤ 3. Z-score | 83 | 13.9% |
| **Female** | | |
| Normal ≤0. Z-score | 123 | 40% |
| Mild ≤1. Z-score | 92 | 31% |
| Moderate ≤2. Z-score | 56 | 18% |
| Severe ≤ 3. Z-score | 30 | 10% |
| **Male** | | |
| Normal ≤0. Z-score | 75 | 25% |
| Mild ≤1. Z-score | 94 | 32% |
| Moderate ≤2. Z-score | 70 | 24% |
| Severe ≤ 3. Z-score | 53 | 18% |

**S2 Table. Data of Z-scores for height for age and weight for age of adolescents of Tharparkar, Sindh, Pakistan.** NRPU 14406.
(XLSX)

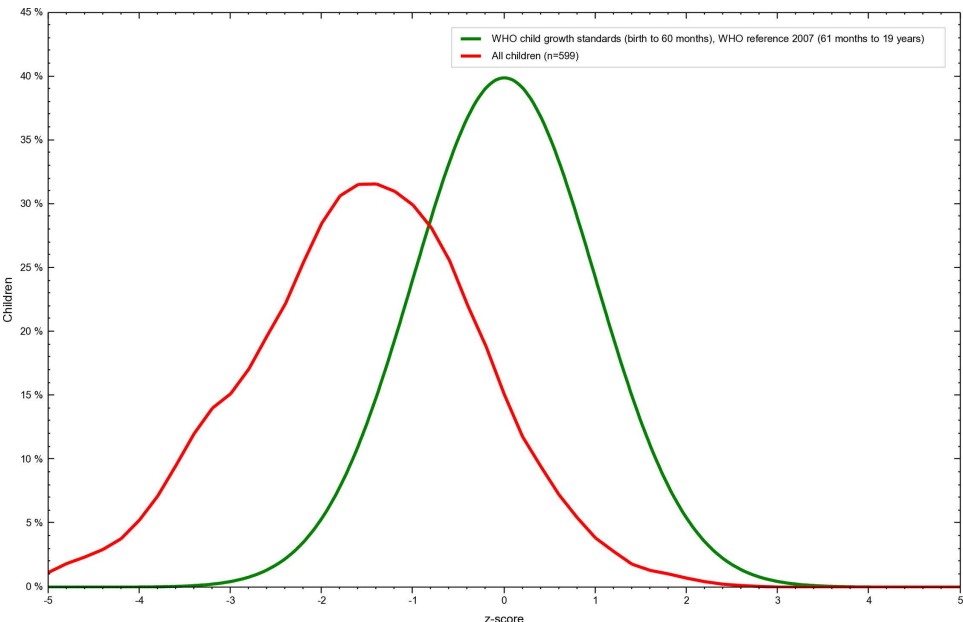

**Fig 3. Overall Z-score distribution of thinness among adolescents of Tharparkar, Sindh, Pakistan.** This graph illustrates the overall Z-score distribution of weight-for-age (thinness) among Adolescents of Tharparkar, Sindh, Pakistan generated using WHO AnthroPlus software. The Z-scores are calculated based on the 2007 WHO Growth Reference, with values below -2 standard deviations indicating stunting. The figure highlights the prevalence of malnutrition in the studied population and visually represents their growth and nutritional status relative to global standards.

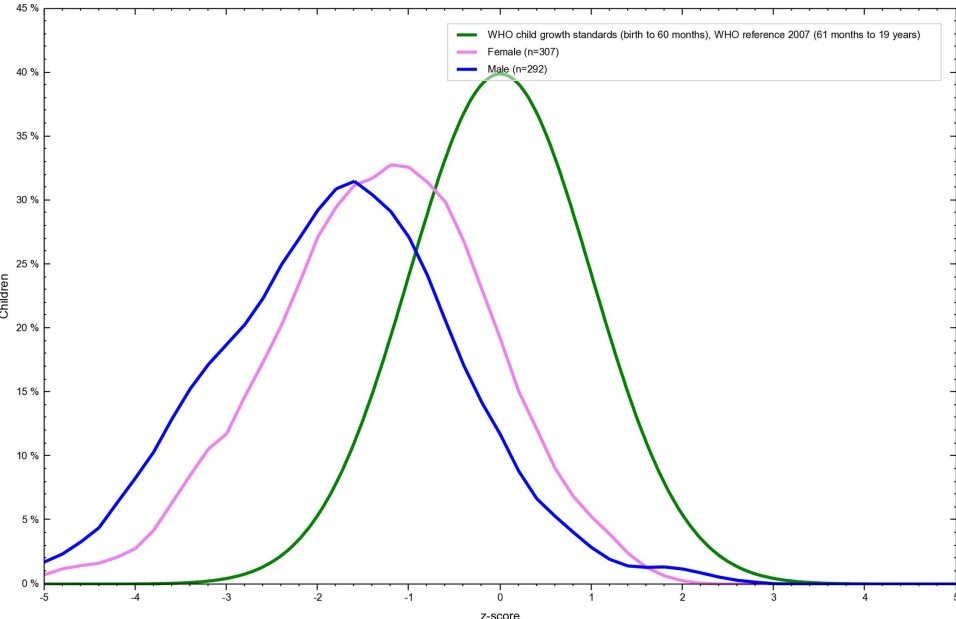

**Fig 4. Gender wise Z-score distribution of thinness among adolescents of Tharparkar, Sindh, Pakistan.** This graph illustrates the gender-wise Z-score distribution of weight-for-age (thinness) among Adolescents of Tharparkar, Sindh, Pakistan generated using WHO AnthroPlus software. The Z-scores are calculated based on the 2007 WHO Growth Reference, with values below -2 standard deviations indicating stunting. The figure highlights the prevalence of malnutrition in the studied population and visually represents their growth and nutritional status relative to global standards.

**Table 8. Association of demographic factors with the prevalence of thinness among adolescents of Tharparkar, Sindh, Pakistan.**

| Sociodemographic Factors | Thinness No | Thinness Yes | Chi-squire | P-value |
|---|---|---|---|---|
| **Age** | | | | |
| Early | 200 (66.9%) | 99 (33.1%) | 2.210 | .331 |
| Middle | 125 (65.8%) | 65 (34.2%) | | |
| Late | 65 (59.1%) | 45 (40.9%) | | |
| **Gender** | | | | |
| Female | 221 (72.0%) | 86 (28.0%) | 13.116 | .0001 |
| Male | 169 (57.9%) | 123 (42.1%) | | |
| **Area** | | | | |
| Rural | 317 (64.7%) | 173 (35.3%) | .204 | .652 |
| Urban | 73 (67.0%) | 36 (33.0%) | | |
| **Education** | | | | |
| Illiterate (0) | 125 (72.7%) | 47 (27.3%) | 12.04 | .007 |
| Primary (1–5) | 97 (67.9%) | 44 (32.1%) | | |
| Secondary (6–10) | 152 (61.3%) | 97 (38.7%) | | |
| Higher Secondary (>10) | 20 (47.6%) | 22 (52.4%) | | |
| **Father's Education** | | | | |
| Illiterate | 167 (64.2%) | 93 (35.8%) | .156 | .693 |
| Literate | 223 (65.8%) | 116 (34.2%) | | |
| **Mother's Education** | | | | |
| Illiterate | 337 (65.3%) | 179 (34.7%) | .067 | .796 |
| Literate | 53 (63.9%) | 30 (36.1%) | | |
| **Socioeconomic status** | | | | |
| Monthly income ≤33000PKR | 329 (65.3%) | 175 (34.7%) | .040 | .841 |
| Monthly income ≥33100PKR | 61 (64.2%) | 34 (35.8%) | | |
| **Siblings** | | | | |
| < 7 | 298 (66.1%) | 153 (33.9%) | .751 | .386 |
| >7. | 92 (62.2%) | 56 (37.8%) | | |
| **Family Members** | | | | |
| < 7 | 155 (68.3%) | 72 (31.7%) | 1.620 | .203 |
| >7 | 235 (63.2%) | 137 (36.8%) | | |

## Acknowledgement

The authors are highly thankful to District Health Officer of Tharparkar at Mithi for cooperation. Z-scores for height for age and Z-score weight for age can be accessed in file S2 Data NRPU 14406.

## Author contributions

**Conceptualization:** Zulfiqar Ali Laghari, Ayaz Ali Samo.

**Data curation:** Aisha Memon, Zulfiqar Ali Laghari, Ayaz Ali Samo.

**Formal analysis:** Aisha Memon, Zulfiqar Ali Laghari, Ayaz Ali Samo.

**Funding acquisition:** Zulfiqar Ali Laghari, Ayaz Ali Samo.

**Investigation:** Zulfiqar Ali Laghari, Ayaz Ali Samo.

**Table 9. Association of dietary factors with prevalence of thinness among adolescents of Tharparkar, Sindh, Pakistan.**

| Dietary Factors | Thinness No =(n) &% | Thinness Yes = (n) &% | Chi-squire | P-value |
|---|---|---|---|---|
| **Lassi** | | | | |
| Infrequent | 307 (65.0%) | 165 (35.0%) | .004 | .948 |
| Frequent | 83 (65.4%) | 44 (34.6%) | | |
| **Yogurt** | | | | |
| Frequent | 82 (63.1%) | 48 (36.9%) | 1.197 | .550 |
| Infrequent | 69 (69.7%) | 30 (30.3%) | | |
| Non frequent | 239 (64.6%) | 131 (35.4%) | | |
| **Milk** | | | | |
| ≤ 1 glass/day | 323 (65.0%) | 174 (35.0%) | .005 | .945 |
| ≥ 2 glass/day | 66 (65.3%) | 35 (34.7%) | | |
| **Butter** | | | | |
| No | 325 (65.9%) | 168 (34.1%) | .813 | .367 |
| Yes | 65 (61.3%) | 41 (38.7%) | | |
| **Chicken** | | | | |
| No | 308 (63.5%) | 177 (36.5%) | 2.884 | .089 |
| Yes | 82 (71.9%) | 32 (28.1%) | | |
| **Egg** | | | | |
| No | 293 (62.7%) | 174 (37.3%) | 5.229 | .022 |
| Yes | 97 (73.5%) | 35 (26.5%) | | |
| **Pulses** | | | | |
| Frequent | 185 (65.4%) | 98 (34.6%) | .016 | .898 |
| Infrequent | 205 (64.9%) | 111 (35.1%) | | |
| **Rice** | | | | |
| Infrequent | 201 (65.7%) | 105 (34.3%) | .092 | .762 |
| Frequent | 189 (64.5%) | 104 (35.5%) | | |
| **Fruit** | | | | |
| No | 133 (64.3%) | 74 (35.7%) | .102 | .749 |
| Yes | 257 (65.6%) | 135 (34.4%) | | |
| **Fresh Vegetables** | 311 (67.6%) 79 (56.8%) | 149 (32.4%) 60 (43.2%) | 5.455 | .020 |
| Frequent | | | | |
| Infrequent | | | | |
| **Preserved Vegetables** | | | | |
| Frequent | 85 (59.9%) | 57 (40.1%) | 2.258 | .133 |
| Infrequent | 305 (66.7%) | 152 (33.3%) | | |

**Methodology:** Zulfiqar Ali Laghari, Ayaz Ali Samo.

**Project administration:** Aisha Memon, Zulfiqar Ali Laghari, Ayaz Ali Samo.

**Resources:** Zulfiqar Ali Laghari, Ayaz Ali Samo.

**Software:** Zulfiqar Ali Laghari, Ayaz Ali Samo.

**Supervision:** Zulfiqar Ali Laghari, Ayaz Ali Samo.

**Validation:** Zulfiqar Ali Laghari, Ayaz Ali Samo.

**Visualization:** Zulfiqar Ali Laghari, Ayaz Ali Samo.

**Table 10. Binary logistic regression analysis thinness adjusted with demographic factors.**

| Variables in the Equation | | | | | | | | | |
|---|---|---|---|---|---|---|---|---|---|
| | | B | S.E. | Wald | df | Sig. | Exp(B) | 95% C.I.for EXP(B) | |
| | | | | | | | | Lower | Upper |
| Step 1ª | Gender (1) | 0.54 | 0.193 | 7.863 | 1 | 0.005 | 1.716 | 1.176 | 2.502 |
| | Area (1) | 0.241 | 0.242 | 0.989 | 1 | 0.32 | 1.272 | 0.792 | 2.045 |
| | Illiterate | | | 5.798 | 3 | 0.122 | | | |
| | Primary Education | -0.83 | 0.395 | 4.427 | 1 | 0.035 | 0.436 | 0.201 | 0.945 |
| | Secondary Education | -0.705 | 0.406 | 3.02 | 1 | 0.082 | 0.494 | 0.223 | 1.094 |
| | Higher Secondary Education | -0.433 | 0.372 | 1.354 | 1 | 0.245 | 0.648 | 0.313 | 1.345 |
| | Family Members (1) | -0.259 | 0.202 | 1.654 | 1 | 0.198 | 0.771 | 0.519 | 1.146 |
| | Siblings (1) | -0.043 | 0.221 | 0.038 | 1 | 0.845 | 0.958 | 0.621 | 1.478 |
| | Early Adolescence | | | 1.464 | 2 | 0.481 | | | |
| | Middle Adolescence | -0.277 | 0.26 | 1.132 | 1 | 0.287 | 0.758 | 0.455 | 1.263 |
| | Late Adolescence | -0.307 | 0.267 | 1.319 | 1 | 0.251 | 0.736 | 0.436 | 1.242 |
| | Constant | -0.164 | 0.388 | 0.179 | 1 | 0.672 | 0.849 | | |

ªVariable(s) entered on step 1: Gender (1), Area (1), Education Level (Illiterate, Primary, Secondary, Higher Secondary), Family Members (1), Siblings (1), Adolescence Stage (Early, Middle, Late)

Outcome **variable**: Thinness status (0 = Not Thin, 1 = Thin)

Predictor **variables**: **Age**: Early, Middle and Late (Reference Category: Late), **Gender**: Male or Female (reference category = Male), **Socioeconomic status**: Categorical, coded as low, middle, high (reference category = High), **Education level**: Primary, Secondary, Higher (reference category = Higher), **Urban/Rural residence**: Place of residence (Urban = 1, Rural = 0; reference =Urban). Adjusted **odds ratio (OR)**: Odds ratio adjusted for age, gender, socioeconomic status, education level, and place of residence. 95**% Confidence Interval (CI)**: The range within which the true odds ratio is expected to fall, with 95% certainty.

**Writing – original draft:** Aisha Memon, Zulfiqar Ali Laghari, Ayaz Ali Samo.

**Writing – review & editing:** Zulfiqar Ali Laghari, Ayaz Ali Samo.

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
