## [Decision Letter · Decision Letter 0]

1 Nov 2024

PONE-D-24-43222Factors and Determinants Associated with Prevalence of Stunting and Thinness among Adolescents of Tharparkar, Sindh, PakistanPLOS ONE

Dear Dr. Samo,

Thank you for submitting your manuscript to PLOS ONE. After careful consideration, we feel that it has merit but does not fully meet PLOS ONE’s publication criteria as it currently stands. Therefore, we invite you to submit a revised version of the manuscript that addresses the points raised during the review process.

We look forward to receiving your revised manuscript.

Kind regards,

Tahir Turk, PhD

Academic Editor

PLOS ONE

Journal Requirements:

2. Thank you for stating the following financial disclosure: "This study was funded by the research grant received from the Higher Education Commission, Islamabad through the National Research Program for Universities (NRPU) project number 14406 for 2021-2024."

3. In the online submission form, you indicated that [Data can be provided by authors only on reasonable request to Principal Investigator on ayazsamo@usindh.edu.pk].

6. We note that there is identifying data in the Supporting Information file <Table S1.xlsx>. Due to the inclusion of these potentially identifying data, we have removed this file from your file inventory. Prior to sharing human research participant data, authors should consult with an ethics committee to ensure data are shared in accordance with participant consent and all applicable local laws.

-Location data

Please remove or anonymize all personal information, ensure that the data shared are in accordance with participant consent, and re-upload a fully anonymized data set. Please note that spreadsheet columns with personal information must be removed and not hidden as all hidden columns will appear in the published file.

Additional Editor Comments:

Currently we have had one reviewer provide feedback on your manuscript with comments as follows:

Reviewers' comments:

Reviewer's Responses to Questions

**Comments to the Author**

1. Is the manuscript technically sound, and do the data support the conclusions?

Reviewer #1: Yes

2. Has the statistical analysis been performed appropriately and rigorously? 

Reviewer #1: No

3. Have the authors made all data underlying the findings in their manuscript fully available?

Reviewer #1: Yes

4. Is the manuscript presented in an intelligible fashion and written in standard English?

Reviewer #1: Yes

5. Review Comments to the Author

Reviewer #1: In this study, the authors examine the factors and determinants linked to the prevalence of stunting and thinness among adolescents in Tharparkar, Sindh, Pakistan. The overall study design and methodology are appropriate for this journal. The authors also report significant associations of stunting and thinness with various risk factors.

Major comments:

1. Were any of the models adjusted for confounding factors? It would be interesting to see the results in the presence of different confounding factors.

2. The criteria for determining education levels are unclear. How was parental literacy assessed? It would be beneficial to explore the association between educational status (years of education) and stunting/thinness.

Minor comments:

1. What do "sibling 1" and "sibling 2" in Table 3 refer to?

2. The manuscript requires thorough English editing.

6. PLOS authors have the option to publish the peer review history of their article (what does this mean? ). If published, this will include your full peer review and any attached files.

**Do you want your identity to be public for this peer review?** For information about this choice, including consent withdrawal, please see our Privacy Policy .

Reviewer #1: No

---

## [Author Response · Author response to Decision Letter 0]

13 Nov 2024

Dear Dr. Turk,

Thank you for the opportunity to revise our manuscript and for the valuable feedback provided by the editorial team and reviewers. We appreciate the thoughtful consideration given to our work and are pleased to hear that our study has been recognized for its merit.

We have carefully reviewed each of the comments raised during the review process. In response, we have made the following changes to enhance the manuscript and address the specific points raised:

Journal Requirements:

Answer# Thank you for your comment. We have updated the reference style as indicated.

2. Thank you for stating the following financial disclosure: "This study was funded by the research grant received from the Higher Education Commission, Islamabad through the National Research Program for Universities (NRPU) project number 14406 for 2021-2024."Please state what role the funders took in the study. If the funders had no role, please state: "The funders had no role in study design, data collection and analysis, decision to publish, or manuscript preparation. “If this statement is not correct you must amend it as needed. Please include this amended Role of Funder statement in your cover letter; we will change the online submission form on your behalf.

Answer# Thank you for your comment. We have modified the statement as suggested and included this information in manuscript revised file.

3. In the online submission form, you indicated that [Data can be provided by authors only on reasonable request to Principal Investigator on ayazsamo@usindh.edu.pk].

Answer of point number 3 and 4# Thank you for your guidance on the data availability policy. We understand and support the journal’s commitment to transparency and accessibility. However, due to its nature, we have some limitations regarding the unrestricted sharing of our data.

Our dataset contains demographic information that could potentially allow for the identification of participants if shared publicly. Additionally, as this is our primary data, we would prefer to share it only with researchers or research groups who agree to acknowledge our contribution by including our team in their subsequent publications. This ensures proper recognition for the extensive work involved in gathering this data while also safeguarding participant privacy.

Therefore, we request an exemption from the open data policy to allow for data sharing under these conditions. We are fully prepared to make the data available upon request to the editorial team for review purposes and to researchers who agree to our stipulations on acknowledgment. We trust this approach aligns with the journal’s standards for accessibility while addressing our concerns about data ownership and participant confidentiality.

Answer: The ethical statement is provided in methodology section.

6. We note that there is identifying data in the Supporting Information file <Table S1.xlsx>. Due to the inclusion of these potentially identifying data, we have removed this file from your file inventory. Prior to sharing human research participant data, authors should consult with an ethics committee to ensure data are shared in accordance with participant consent and all applicable local laws.

-Location data

Please remove or anonymize all personal information, ensure that the data shared are in accordance with participant consent, and re-upload a fully anonymized data set. Please note that spreadsheet columns with personal information must be removed and not hidden as all hidden columns will appear in the published file.

Additional Editor Comments:

Currently we have had one reviewer provide feedback on your manuscript with comments as follows:

Reviewers' comments:

Reviewer's Responses to Questions

Comments to the Author

1. Is the manuscript technically sound, and do the data support the conclusions?

Reviewer #1: Yes

Answer# Thankful to reviewer for feedback.

2. Has the statistical analysis been performed appropriately and rigorously?

Reviewer #1: No

Answer# Thankful to reviewer for feedback. We have added additional statistical analysis as suggested by reviewers.

3. Have the authors made all data underlying the findings in their manuscript fully available?

Reviewer #1: Yes

Answer# Thankful to reviewer for feedback.

4. Is the manuscript presented in an intelligible fashion and written in standard English?

Reviewer #1: Yes

Answer# Thankful to reviewer for feedback.

5. Review Comments to the Author

Reviewer #1: In this study, the authors examine the factors and determinants linked to the prevalence of stunting and thinness among adolescents in Tharparkar, Sindh, Pakistan. The overall study design and methodology are appropriate for this journal. The authors also report significant associations of stunting and thinness with various risk factors.

Major comments:

1. Were any of the models adjusted for confounding factors? It would be interesting to see the results in the presence of different confounding factors.

Answer# We are very thankful to the reviewer for this comment. We have adjusted the stunting and thinness based on the demographic factors we have studied. A binary logistic regression model examined the association between stunting and demographic factors. Primary and higher secondary education levels and middle adolescence stage are significantly associated with reduced odds of stunting (Table 6). Males are at higher odds for thinness, while individuals with primary education have significantly lower odds of thinness compared to those who are illiterate (Table 10).

2. The criteria for determining education levels are unclear. How was parental literacy assessed? It would be beneficial to explore the association between educational status (years of education) and stunting/thinness.

Answer# We are grateful to the reviewer for this comment. Education was determined by 0 means illiterate, 1-5 means primary education, 6-10 means secondary education, and >10 means higher secondary education. We have explored the association between educational status and stunting and thinness. Please follow Chi-square and logistic regression analysis tables.

Minor comments:

1. What do "sibling 1" and "sibling 2" in Table 3 refer to?

Answer# We are very thankful to the reviewer for pointing out this mistake. Siblings have been categorized into 2 groups sibling 1 indicates the number of siblings <7 and Sibling 2 indicates the number of siblings >7

2. The manuscript requires thorough English editing.

Answer# We have modified the language where needed.

---

## [Decision Letter · Decision Letter 1]

10 Jan 2025

Factors and Determinants Associated with Prevalence of Stunting and Thinness among Adolescents of Tharparkar, Sindh, Pakistan: A community based study.

PONE-D-24-43222R1

Dear Dr. Samo,

We’re pleased to inform you that your manuscript has been judged scientifically suitable for publication and will be formally accepted for publication once it meets all outstanding technical requirements.

Kind regards,

Tahir Turk, PhD

Academic Editor

PLOS ONE

Reviewers' comments:

Reviewer's Responses to Questions

**Comments to the Author**

1. If the authors have adequately addressed your comments raised in a previous round of review and you feel that this manuscript is now acceptable for publication, you may indicate that here to bypass the “Comments to the Author” section, enter your conflict of interest statement in the “Confidential to Editor” section, and submit your "Accept" recommendation.

Reviewer #1: All comments have been addressed

2. Is the manuscript technically sound, and do the data support the conclusions?

Reviewer #1: Yes

3. Has the statistical analysis been performed appropriately and rigorously? 

Reviewer #1: Yes

4. Have the authors made all data underlying the findings in their manuscript fully available?

Reviewer #1: Yes

5. Is the manuscript presented in an intelligible fashion and written in standard English?

Reviewer #1: Yes

6. Review Comments to the Author

Reviewer #1: The authors effectively addressed all of my concerns. The quality of the figures could be enhanced, and there are still some typos in the manuscript that need the authors' attention.

---

## [Editor Report · Acceptance letter]

PONE-D-24-43222R1

PLOS ONE

Dear Dr. Samo,

I'm pleased to inform you that your manuscript has been deemed suitable for publication in PLOS ONE. Congratulations! Your manuscript is now being handed over to our production team.

Kind regards,

on behalf of

Dr. Tahir Turk

Academic Editor

PLOS ONE